# Periurban Areas in the Design of Supra-Municipal Strategies for Urban Green Infrastructures

Eva Fernández-Pablos [1], Amparo Verdú-Vázquez [2,*], Óscar López-Zaldívar [2] and Rafael V. Lozano-Diez [2]

1 Técnico Responsable del Mantenimiento de las Huertas-Jardín del Palacio de Boadilla del Monte, 28660 Madrid, Spain; evafdzpablos@gmail.com
2 Departamento de Tecnología de la Edificación, Escuela Técnica Superior de Edificación, Universidad Politécnica de Madrid, 28040 Madrid, Spain; oscar.lopezz@upm.es (Ó.L.-Z.); rafaelvicente.lozano@upm.es (R.V.L.-D.)
* Correspondence: amparo.verdu@upm.es; Tel.: +34-9133-6760-5

**Abstract:** Nowadays, an increasing number of large cities, districts, and towns have tools for the Planning and Management of Green Infrastructures. All such tools seek a progression towards a future city model that is more resilient on an environmental, economic, and social level. To achieve this, emphasis is placed on the creation of a green infrastructure and, particularly, on improving urban biodiversity, urban forests, the value of natural areas in the urban environment, periurban agriculture, ecological connectivity, and accessibility. Moreover, the recent COVID19 health crisis has further highlighted that the city dweller's relationship with the environment requires a reconciliation with nature and rural life that goes beyond typical compartmentalization. The objective must be to emphasize the need to establish creative processes which, through micro-scale activities (landscaping), generate the articulation of visible actions on a territorial scale (landscape planning) in both the natural environment (environmental landscape planning) and the urban environment (town planning based on the landscape). This article analyzes the issue of the large towns in south-west Madrid, where there is a dramatic divide on the border between the city landscape and the surrounding natural or agricultural landscape, and where there is an increasing need to establish landscapes with a certain uniqueness and to classify them as protected periurban areas, nature reserves, or land for which use and management is regulated. It is therefore important to develop environmental quality standards to assess Green Infrastructures as a whole: the administrative processes, their design, construction, maintenance, and resilience. This research focuses upon how this change in the planning and management of green periurban areas improves the multifunctionality of periurban spaces along with the intrinsic quality of the landscape, and promotes the city's sustainability and resilience and improves governance. From the conclusions drawn, it should be noted that analysis, design, and action should be built on premises of sustainability and multifunctionality, and comply with the criteria for characterizing elements as green infrastructure. In the field of study, the characterization of the periurban area, and its subsequent assessment as a green infrastructure, provide the guidelines for action for devising an Open Space Strategy. This strategy constitutes a cross-disciplinary planning tool for local authorities when reading the landscape.

**Keywords:** periurban parks; green infrastructure; urban ecology; resilience; urban forest; sustainable urban strategies

## 1. Introduction

Green Infrastructure (GI) has become a top issue for planning and policy-making processes in fields such as urban development, environmental conservation, agriculture, and forestry [1]. Sustainable planning and management of cities requires the integration of Green Infrastructure in local and regional government, not only as a design tool, but also for assessment and monitoring. The benefits that Green Infrastructure (GI) has for human well-being are fully recognized, but the means for its systematic application are still

scarce. The identification, fostering, and maintenance of a well-planned green infrastructure network can offer our society significant environmental, economic, and social benefits [2]. It has also been highlighted as a powerful tool for helping cities to adapt to climate change by providing numerous ecosystem services [3].

Local authorities play a key role in the maintenance of green infrastructure at a local level and the formal recognition of this function in the budgetary systems would promote its inclusion in their agenda [4]. The implementation of a green infrastructure requires both time and spatial planning, and a design that is incorporated in territorial planning processes. Furthermore, its implementation requires the cross-disciplinary collaboration and cooperation of multiple agents such as politicians, managers, social organizations and the general public as a whole [5]. Both Green Infrastructure and Ecosystem Services are characterized by their multifunctionality and cross-disciplinary nature, enabling them to be used to devise city sustainability strategies for green policies, mobility, citizen well-being, and public health, among others [6].

The application of the GI development agenda in Europe requires the worldwide development and implementation of green infrastructure plans, although it is recognized that there are key differences between countries, particularly in relation to climate, social characteristics, institutions, and landscape planning frameworks [7].

For the European Commission, the following facts support the need to promote a green infrastructure in the Member States [8]:

- The landscape of Europe is increasingly fragmented;
- Wildlife must have the chance to exist outside of protected areas;
- A green infrastructure helps to maintain the valuable services provided by the ecosystems;
- We must make space for nature by adopting a more comprehensive approach to land use;
- Appropriate territorial planning helps to create a green infrastructure;
- EU financial instruments may be used to promote the development of a green infrastructure;
- The formulation of a European strategy for the creation of a green infrastructure throughout Europe as the objective of the 2020 Biodiversity Strategy.

Spain's Green Infrastructure Strategy was approved in October 2020 [9], which is a framework tool to assist the government, the autonomous regions, and the local authorities in developing plans, programs, and actions aimed at strengthening the connectivity of the ecosystems, at mitigating the effect of climate change and at reinforcing ecosystem services within the scope of their competences.

Nowadays, an increasing number of regions, large cities, and towns have Green Infrastructure Planning and Management instruments aimed at boosting multifunctionality and improving biodiversity. They all seek to achieve a more environmentally, economically, and socially resilient future city model and, as such, the focus is on the creation of a green infrastructure, placing the emphasis on improving ecosystem services, on enhancing urban biodiversity, on the management of the urban forest, on the value of natural areas in the urban environment, on farming for the local market, on ecological connectivity, and on accessibility [10].

In the local context, it is worth highlighting two recently published studies, which focus upon the analysis of urban green belt management indicators and a proposal for guidelines for local authority decision-makers when developing and implementing their strategies: the "Analysis of the Conservation of the Urban Green Infrastructure in Spain" [11] and the Municipal Green Infrastructure Guide [12].

In the field closest to this study, the Green Infrastructure and Biodiversity Plan for the city of Madrid, drafted in 2018 [13], includes biodiversity as part of the strategic planning of the green infrastructure. It considers sustainability, connectivity, increasing the degree of connection between the green spaces located inside and outside the city, and climate change, contributing towards mitigating its effects and having a mass of vegetation

capable of absorbing the largest possible amount of emissions. It also looks at boosting the permeability, water retention, and biological properties of the soil, through implementing sustainable urban drainage techniques (Department of the Environment and Mobility, Madrid City Council, 2018) [13].

This research focuses upon the issue of the large towns in south-west Madrid, where there is a severe fracture on the border between the city and surrounding natural or agricultural landscape, and where it is increasingly necessary to identify areas with a certain uniqueness and classify them as protected periurban spaces, nature reserves, or sites for which use and management is regulated. The location of these areas is key to the implementation of ecological networks in the city, given the need to restore and maintain the functional connection between natural and urban spaces, as they are the only resource for creating green strategies which improve the overall quality of the urban environment and the lives of its residents [1].

The uninterrupted urban landscape offered by these large cities makes it necessary to think about green infrastructures on a supra-municipal scale, with periurban spaces as their natural connection [14]. Their characterization and evaluation are intended to:

1. Promote the landscape linkage of large green areas on the outskirts of large cities with the structures of landscape value in which they are incorporated and which either retain or should be given value, such as the agricultural areas, protected natural areas, livestock trails, drovers' routes, traditional paths, heritage elements, etc.;
2. Maintain, improve, and extend the networks of periurban green areas in south-west Madrid, both between them and with the urban green spaces network;
3. Promote sustainable mobility, accessibility, and the communication of these green areas with urban areas, prioritizing sustainable transport proposals;
4. Acknowledge the need to assess and diagnose the quality of the green areas in order to promote quality landscapes in the urban and periurban environment of south-west Madrid;
5. Devise a strategy as an instrument of governance, due to its supra-municipal nature, and a tool to evaluate future actions;
6. "The creation of a work and collaboration network between the various municipal strategies, making recommendations to the planners and designers concerning the quality of the spaces and the environmental education required by the community in order to ensure the correct use of the green areas" [15].

This approach to green infrastructure planning on a supra-municipal scale requires a dynamic perspective in which design criteria must promote this evolution and change in a way that is compatible with the maintenance of the space. Studies by McHarg, Michael Hough, and John Tillman can be considered to be the precursors of the need for design to incorporate originality and creativity whilst recovering natural shapes and processes [16].

In the search for balance between design, creativity, and ecology, Nina Marie-Lister describes three aspects in which ecological design can offer opportunities for sustainability, planning, management, and maintenance [16]: firstly, a detailed understanding of the ecological systems from which an adaptive modeling process is derived; secondly, ecological design should be adaptable, inclusive, and resilient; finally, all of the above leads to a process which enables a connection with the social dimension, and it must reflect the cultural-natural connection that is fundamental to green spaces.

Adaptive design is an important tool in the learning process which leads to the creation of large green spaces. If the various necessary agents are involved, it is possible to find a balance between nature and culture, essential to the future of urban and periurban green areas, since only through the appropriation of space is care generated.

Regarding resilience, in landscaping and, in particular, in large green areas and periurban spaces, a key issue is the maintenance of the pressure on systems that have opposing properties: they must be flexible, adaptable, socially dynamic, and revive places that are missing in the city [17].

Meanwhile, Jack Ahern identifies five approaches for resilience-based design [18]: multifunctionality, diversification to improve the capacity to respond not only to natural changes but also to socio-economic changes, interconnection of work scales, adaptive design as a laboratory of ideas to create dynamic responses, and analysis of the implementation of regulations or new policies, the need for which has become evident during the current COVID 19 crisis [19].

In addition, this health crisis has further highlighted that the city dweller's relationship with the environment requires a reconciliation with nature and rural life that goes beyond typical compartmentalization and fragmentation. The need for a healthier lifestyle and the time constraints that prevent daily access to natural surroundings close to urban areas makes it necessary for many people to look for spaces close to where they live in order to do sport and healthy leisure activities, spaces in which they can reconnect with nature through urban agriculture or routes to escape the city and head out to the periurban environment of their communities [20]. These functions are allocated to the periurban parks of south-west Madrid [21].

Society is demanding new ways of existing within the city, spaces to connect with the natural environment and urban nature alternatives which emerge not only from personal or group initiatives but also from the local authorities [22].

## 2. Methodology

This research is a response to the guidelines set out in the European Landscape Convention [23], recognizing the fundamental role of identifying, characterizing, and classifying the landscapes in the preliminary phase of any landscape policy, the formulation of related landscape strategies within the scope of their competences, the integration of the landscape in territorial planning policies, be they general or sectoral, and the exchange of information, methodologies, and experiences.

The area of study is defined territorially by the basins of the three streams which structure the landscape and generate the periurban green areas analyzed in the case study: the Culebro stream, the Butarque stream and the El Soto stream, in the municipal districts of Móstoles, Leganés, Fuenlabrada and Alcorcón, whose principal periurban green areas have been the subject of a detailed study due to their importance as a periurban green infrastructure.

The choice of this territorial area is a response to the need to protect the periurban space as a conservation area between the large towns which make up south-west Madrid. Since the seventies, the urban growth in the towns of south-west Madrid has been continuous and progressive, and both their expansion and allocation of infrastructures and services have occurred in line with a production model based on the consumption of resources and available land which has meant that, on a local scale, urban green spaces have become fragmented due to an absence of planning, occupying the spaces which remain in this constant growth or "among" future urban developments.

On a regional scale, south-west Madrid functions as a bridge area between the continuation of the metropolitan area formed by the large towns in the west and south of the Madrid region and smaller towns which, over the last few years and recently, with the health crisis, are experiencing intense growth linked to the search for other lifestyles featuring a greater connection with nature.

In this context, the river and stream courses constitute, together with urban forests, the essential elements for connecting the city with its environment [21] and, in this research, they underpin the definition of the physical environment of the work, based on which an Open Space Strategy will be defined for south-west Madrid by means of the multifunctional assessment of its periurban green areas.

Thus, based on the analysis of the connection options between urban and natural environments, and the characterization of periurban spaces, a multi-scale research methodology is applied to the design of open space strategies in the south-west of the Madrid region [2]. This methodology combines the level of detail in the characterization of peri-

urban green spaces and green areas with the definition of objectives and actions needed in a supra-municipal strategy. To this end, supra-municipal planning tools are used and synergies are sought between the functions attributed to a landscape components to form part of an Open Space and Green Infrastructure Strategy. These functions are based on the references collected by the European Environment Agency for Green Infrastructures [24] and the Open Space Strategy of London [25] for the evaluation of open spaces. Finally, they are classified on the basis of the multifunctionality attributed to an element defined as a green infrastructure: environmental, socio-educational, and economic

### 2.1. Tools for Supra-Municipal Assessment and Planning

In order to deal with studying the urban and periurban green areas from the perspective of multifunctionality, connectivity, and reservoirs in a large town environment, the design of Open Space and Green Infrastructure strategies offers the principles used to develop new tendencies in strategic planning tools as a planning proposal which includes them and reflects them in development projects.

#### 2.1.1. Green Infrastructure Strategies

Green infrastructures represents a modern vision of landscape management and, in particular, of the handling of social, economic, and natural resources in the urban- and periurban environment, by providing a long-term sustainable research and resource management strategy [26]. They provide designers, politicians, and citizens with the option to reconsider their approach to their environment.

For the European Environmental Agency, green infrastructure is the appropriate concept to enable us to get closer to the connectivity of ecosystems. Likewise, it promotes its integration into land-use planning through the assessment of multifunctional zones and the incorporation of habitat recovery measures and other connectivity aspects into policies and plans [27].

The establishment of a green infrastructure urban plan also offers the possibility of a multifunctional diagnosis of the area, and a rating of the quality of the urban green areas and of the natural environment, helping to make decisions about the actions to implement. In relation to the periurban area, green infrastructures, as a project on the urban-natural border, contribute towards improving the perception of these places by valuing the landscape and the provision of resources to the population, which are fundamentally social and for improving environmental quality [2].

The administrative and financial aspects of green infrastructures should focus on the integration and coordination of policies addressing social, economic, and environmental issues and promote cross-administrative dialogue, which boosts territorial cohesion.

#### 2.1.2. Open Space Strategies

At a local level, Open Space Strategies are considered to be a cross-disciplinary planning instrument as they affect all areas of a city's government: environment, town planning, health, education, and transport. All of them must be grouped in the fields of diagnosis, landscape design, or restoration of green spaces [28].

The aim of these strategies is the creation and protection of a network of quality open spaces through which residents can improve their perception of the city, foster urban regeneration, environmental and economic revaluation, the accessibility of clean recreational areas, and proposals for public use which are accessible to everyone, all in accordance with the principles of universality. They should be implemented by means of projects that enhance biodiversity and the implementation of rainwater harvesting solutions, with particular emphasis on physical, functional, and social availability and the adoption of measures to reduce the impact of climate change by incorporating monitoring procedures in the conservation and maintenance of green spaces.

Open Space Strategies improve connectivity and synergy, and increase a neighborhood's opportunities to share spaces, revitalize them, and attract investment and trade.

This connection boosts the use of the spaces by increasing the potential population that can access them from other towns [29].

Regarding the environment, the possibility of transferring natural vegetation from their original zones also increases. However, this may also be the vehicle which enables invasive species to enter or pests and diseases to spread, and it is therefore important to consider these two aspects in their design and maintenance. The functions are comparable to those of green infrastructure, as the open space network is a key element in creating a grid on which to build the green infrastructure.

In periurban areas, the differentiating element is found when reaching urban areas where green infrastructure involves the integration and cross-disciplinary nature of the measures to be implemented, including those which involve landscape-informed planning decisions. Meanwhile, Open Space Strategies are specific to local levels, while Green Infrastructure Strategies can be applied on a supra-municipal or regional scale [30].

In the area of study, the application of this methodology seeks to improve ecological and functional connectivity, fundamental elements being the network of periurban green areas, their connection to the urban green infrastructure and to the Network of Protected Natural Areas, the landscape integration of the main ecological corridors, and the promotion of sustainable mobility, all within the context of improving the quality of citizens' lives.

These periurban green areas function as core spaces which give rise to the development of Open Space Strategies between the periurban area and the city. In this analysis, green areas are selected which, due to their location, ecological function, surface area, and public use program, are key to the development of multifunctional proposals linked to urban areas.

### 2.2. Research Methodology

The theoretical principles underpinning this research have been set out using a landscape ecology environmental landscape planning and green city planning rationale. Meanwhile, the theoretical principles of the study of periurban spaces were outlined in terms of multifunctionality, connectivity, and reservoirs in the big town environment, and the principles according to which new trends in planning tools are developed were explained using the creation of open space strategies and green infrastructures as a planning proposal which includes them in development plans.

The characterization of the periurban landscape in the study area is based on the theoretical foundations of landscape ecology, which seeks to establish the spatial structure of the landscape, that is, the distribution patterns of its elements, as well as the functioning of this spatial structure and its evolution over time. Through the historical analysis of aerial images, cartographic bases, and municipal management plans, eight categories are established, which are also defined as elements potentially generating green infrastructure. This process has been contrasted with a field study. The categories are: characterization of the urban environment, landscape units, areas of landscaping interest, agricultural landscapes, green areas, and green and ecological corridors: hydrology, paths, and livestock trails.

The methodology applied to the present research proposes a working method based on the integration of methods for the identification and assessment of the green belt zones and supra-municipal and strategic planning tools. The resulting matrix makes it possible to diagnose green and periurban natural area projects by assessing their potential as a green infrastructure within the framework of Open Space Strategies (Figure 1). The conclusions of this study have made it possible to develop the specific proposals of the Strategy with the definition of Objectives and Actions and to draft Guidelines in common areas of action within the territory.

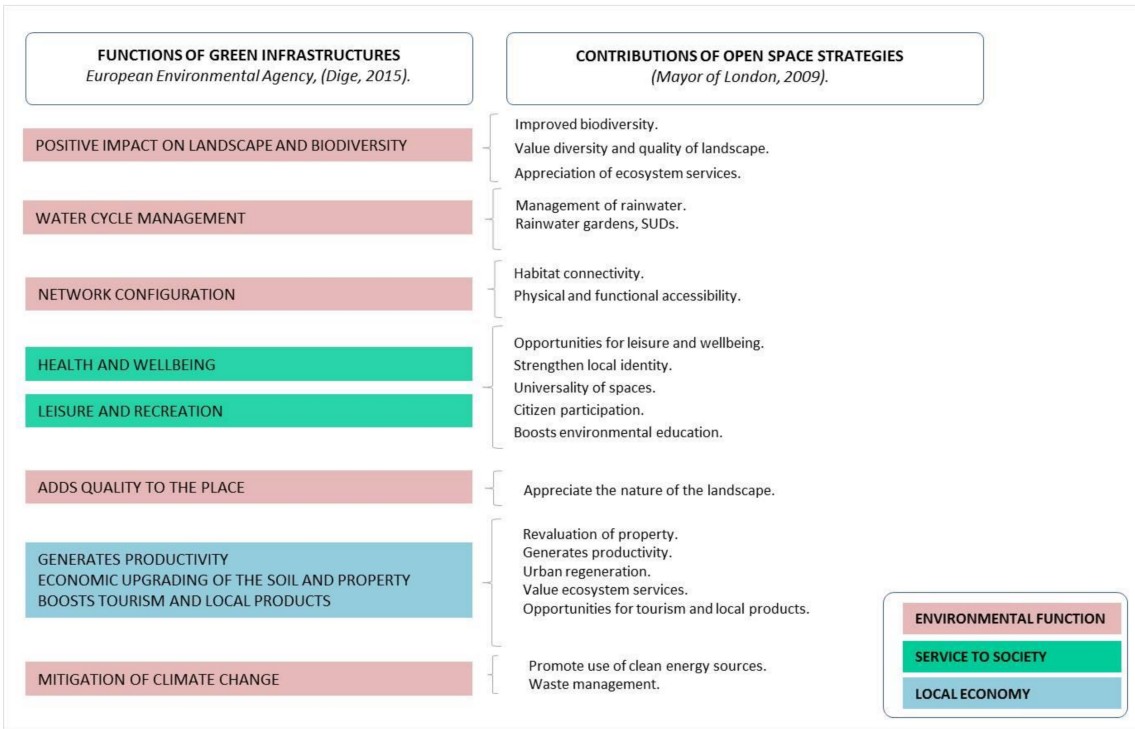

**Figure 1.** References in the development of multifunctional criteria for the evaluation of periurban space [2].

## 3. Results

The application of the methodology in the study area has resulted in the definition of existing and potential elements of green infrastructure, as well as three connecting axes (Figure 2). Based on this master plan, three belts are structured, through which the south-west of Madrid will have a network of spaces that will contribute to improving the quality of life and well-being of citizens, and the resilience of cities and their governance. (Figures 2–6). These belts are:

- "Northern strategy", defined as covering the green area formed by Presillas Park-Butarque Stream and its connection with the areas of landscaping interest north of Alcorcón towards the Public Utility Mount in Boadilla; with the green spaces network and Madrid circular cycle path in Carabanchel; towards the south with the Polvoranca Open Space Strategy;

- "Southern strategy", defined as covering between Bosquesur, the Fuenlabrada Agricultural Park, and the Regional Park in south-east Madrid. It connects with the northern strategy via Polvoranca Park;

- "Western strategy", based on the creation of a belt between the areas of landscaping interest in Villaviciosa de Odón and the Fuenlabrada Agricultural Park; its core area is the redefinition of the Móstoles Green Network as a potential Green Infrastructure Network and the value of the Regional Park of the Middle Course of the Guadarrama River.

The strategy enables a connection between the rural areas of the west of the region via the Regional Park of the Middle Course of the Guadarrama River, with the Regional Park in the south-east, with the Madrid Green Infrastructure Network in the Casa de Campo in the north, and with the linear park along the River Manzanares.

The graphical representation shows a detailed map of each strategy followed by the structured definition of each objective and the actions proposed (Figures 2–6). The classification of the actions for each objective has been carried out based on the environmental, social, and economic functions. By way of summary, these lines of action have been iden-

tified in Tables 1–3, the result of performing an analysis of weaknesses and potential in environmental, social, and economic aspects:

**Table 1.** Environmental Function analysis.

| Potential | Weaknesses |
| --- | --- |
| The connectivity of habitats and improvement of biodiversity is one of the factors offering the biggest chance of improvement in the context of a strategy for the majority of the elements characterized in periurban areas and corridors. Areas of natural value are a source of biodiversity, but actions for improvement are required given that their potential is linked to the intrinsic value as a Protected Natural Area, and there is a strong conflict which threatens the fragility of this landscape due to its proximity to the large towns. It is important to highlight the importance of periurban areas used for individual or shared recreational and vegetable plots which, unlike traditional agricultural areas, offer great potential in the recovery of environmental functions. In the complex fabric of the periurban landscape of south-west Madrid, the river corridors and green corridors, be they livestock trails, green routes, or linear green spaces, must serve as defining elements of the Open Space Strategy due to their importance in network configuration, habitat connectivity and, in general, their positive contribution to the landscape and the recovery of the area's natural values. In addition, it is important to highlight the areas associated with linear infrastructures: public transport, roads, and highways, as well as the recovery of traditional farm paths in order to finish configuring the physical landscape connections. It is about boosting the possibilities offered by a consolidated infrastructure network by offering proposals concerning the existing actions: underpasses, footbridges, rights of way, embankment treatment, and interchanges. | In relation to the management of the water cycle and mitigation of climate change, the existence of systems to manage storm water and waste water are common, particularly in the areas closest to urban areas or industrial or commercial activities. These are unquestionable procedures in urban systems, but on analyzing the existence of elements related to sustainability such as the incorporation of clean energies or sustainable water cycle management, we find the opposite situation, which demonstrates the need to design and plan thinking about the multifuntionality of the element, going beyond offering solutions which only respond to one need. The incorporation of technologies for the reuse or use of storm water or runoff water is a widespread deficiency in the structures assessed. The accessibility between these spaces is a handicap due to the presence of a strong grey infrastructure network with large junctions and intersections which hinder both environmental connectivity and citizen mobility. The periurban agricultural belt is largely unappreciated and it is subjected to significant urban pressure, the construction of infrastructures and the valuation of its owners. In this area of research they are structures subject to conflicts of interest which would require a Special Plan to organize their use and boost their financial, environmental and social profitability. |

**Table 2.** Social Function analysis.

| Potential | Weaknesses |
|---|---|
| The elements which are more established and socially entrenched, where the main objective is to promote social and educational participation, are those with the greatest potential for achieving the social function and these are the spaces where there are vegetable gardens and educational establishments: primary schools, secondary schools, and environmental education centers.<br>Most of the actions, projects, plans, etc. carried out in the environment of the large towns of south-west Madrid have a clear objective: to encourage leisure activities and well-being among the population, be it through consumption-leisure or leisure activities related to nature, health, or sport. For this reason, there are numerous amenities, structures, and facilities allocated to this purpose, the majority of them located in periurban areas, making them potential green infrastructures. The open space strategy will promote the mitigation of the "heat island effect" in cities and provide new opportunities for doing sport in spaces which are closer to the citizens. The effect on public health can be researched and assessed with the aim of quantifying the benefits of the creation of an open space network for the population of south-west Madrid.<br>The improvement of the connectivity that this strategy provides will enable the planning of projects for safe school paths, ensuring safe movement through places which are attractive and beneficial to children. | In the large towns included in the scope of the research, it is difficult to find cultural, historical, or patrimonial references on which to base the identity of the place. A study of the new urban cultures is necessary in order to design networks of spaces which satisfy their needs, those of multicultural populations which will make up the city of the future. Without any doubt, encouraging citizens' participation is an outstanding issue as mentioned above. As such, the strategy must include their active and continued participation in decision making and in the future development of proposals. With regard to environmental education, the Environmental Education Centers at Polvoranca Park and the Bosquesur carry out a very important educational program in order to make their environment known, unlike Móstoles whose center has a program limited to the urban environment. In any case, it does not cover other educational elements present in the town beyond its scope of action, normally the very space in which it is located. The programs should be incorporated in the education system of the towns in a creative manner, responding to the schools' needs.<br>The implementation of educational innovation programs to improve biodiversity in schools is currently a great opportunity to achieve group diversification in different spaces whilst improving the open spaces at schools. The Environmental Education Centers should offer their experience as teachers and environmental professionals in order to achieve these objectives in a joint project with the education community. |

**Table 3.** Economic Function analysis.

| Potential | Weaknesses |
|---|---|
| The elements of which a green infrastructure is composed must contribute to urban regeneration and economic revaluation, as well as enhancing ecosystem services. The complexity of the landscape in the area of study favors the presence of structures which already contribute to this economic function and which will see this value boosted in the context of the Open Space Strategy, such as agricultural, commercial, and industrial areas in the context of a green infrastructure network.<br>The structures with the greatest potential in the periurban area include vegetable gardens, agricultural areas, and the Regional Park of the Middle Course of the Guadarrama River, due to its potential for boosting tourism and contributing towards urban regeneration and land revaluation.<br>The green corridors contribute to the revaluation of land, property, and commercial and sports areas; they boost tourism, and contribute to the local economy as facilities with high public use due to their quality and diversity. | The urban and periurban green belt in south-west Madrid has been experiencing a situation of continued decline and downturn since the 2008 economic crisis, which is reflected by a significant deterioration in both the plant elements and in its infrastructure and facilities. The investments needed to reverse this situation are increasingly high and require innovative solutions, changes to design criteria, and maintenance needs which have been adjusted in line with the current local investment options.<br>The periurban agricultural landscapes of south-west Madrid are, for the most part, characterized by being plots with low productivity, except for Fuenlabrada Agricultural Park. The significance of this situation may lie precisely in this weakness, since the incorporation of agri-environmental measures may represent a consensual solution between the owners and participation in the Green Infrastructures Strategy. |

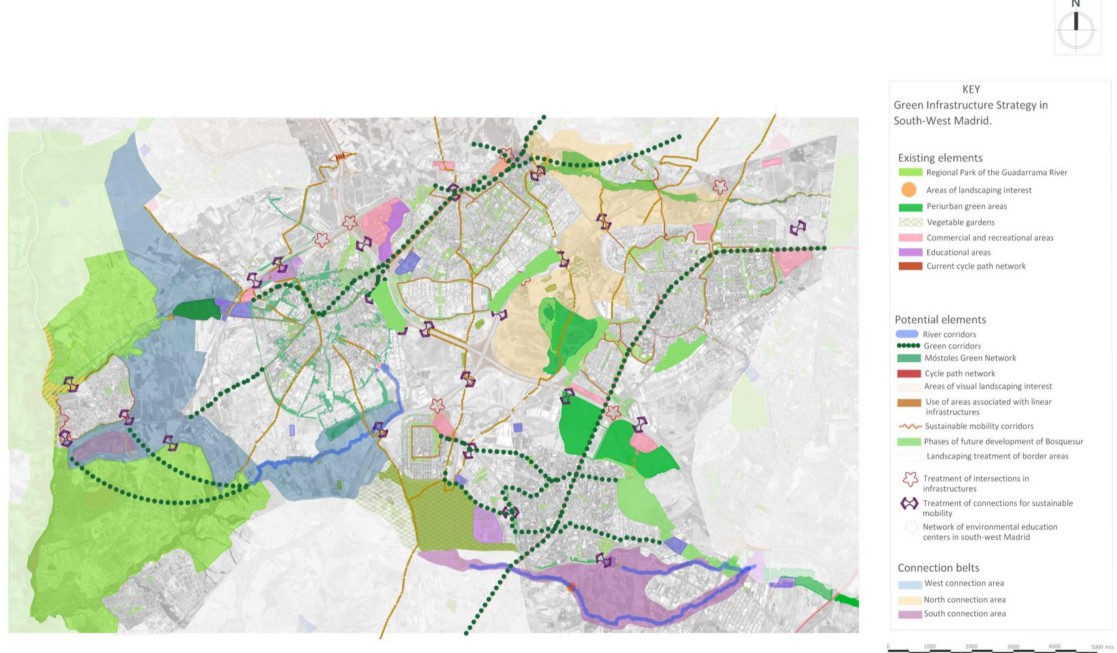

**Figure 2.** Overview map. Prepared by the authors.

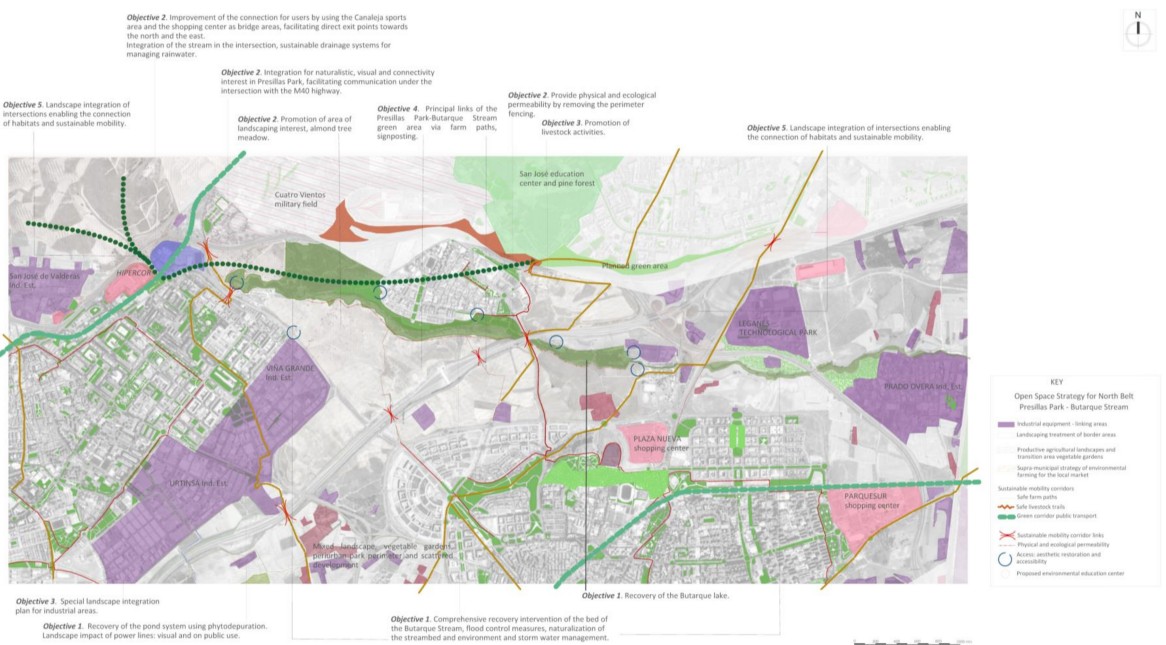

**Figure 3.** Northern strategy a.

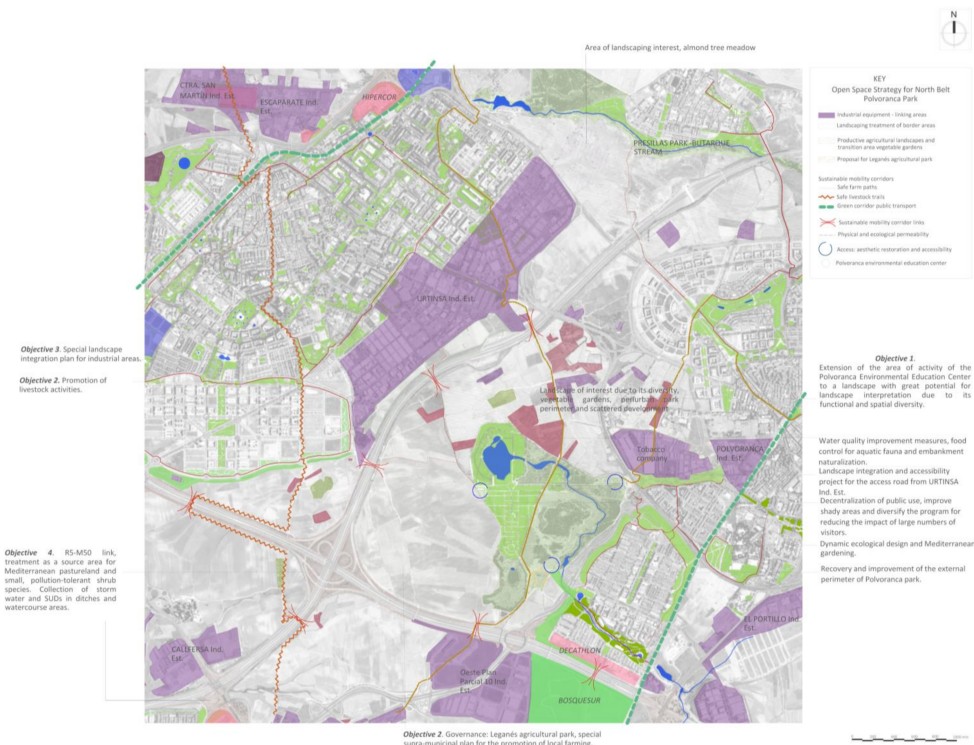

**Figure 4.** Northern strategy b.

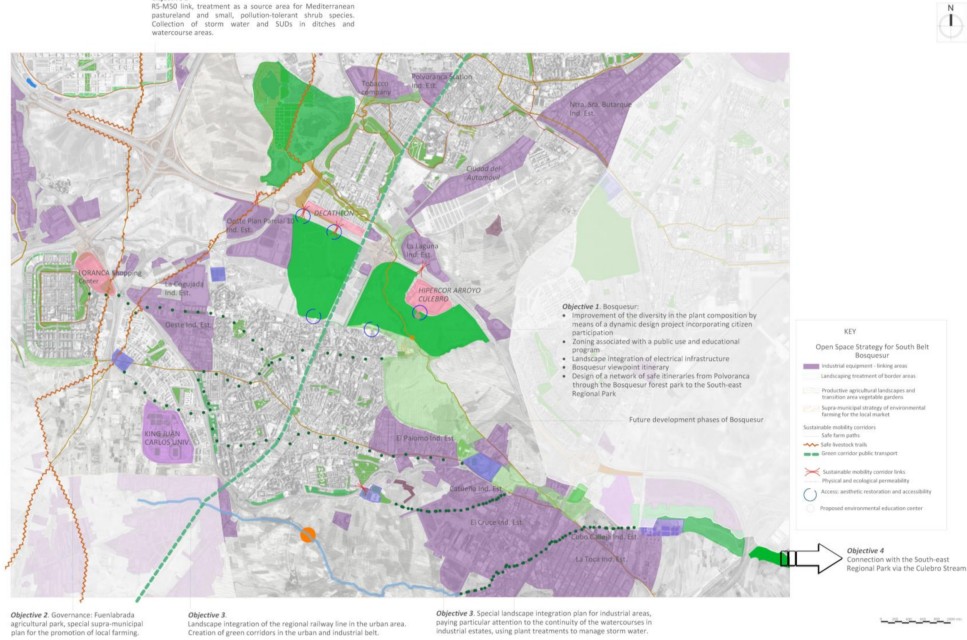

**Figure 5.** Southern strategy.

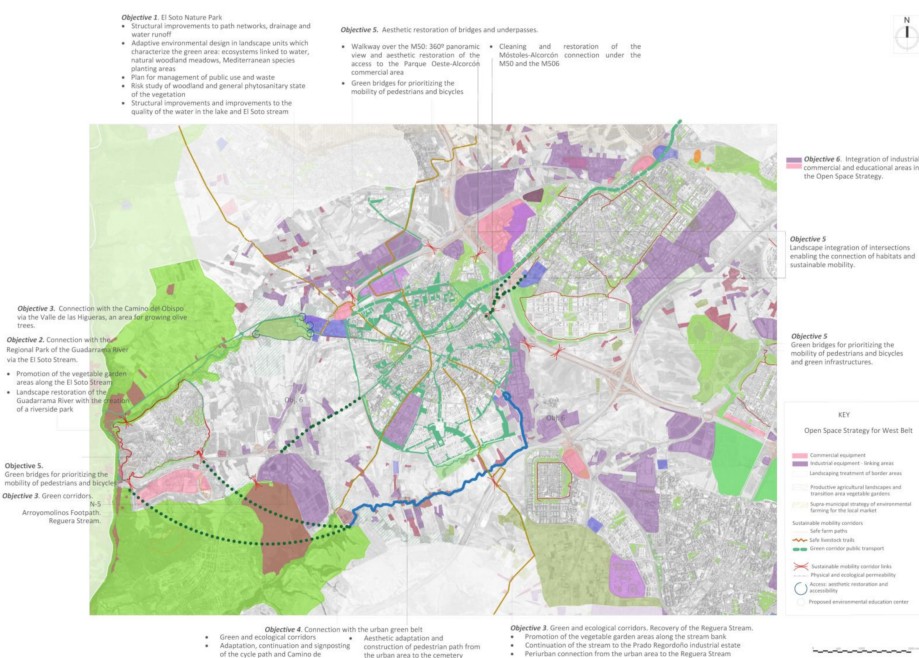

**Figure 6.** Western strategy.

## 4. Conclusions

Nowadays, the design of Green Infrastructure Strategies is an essential pillar and serves as a multifunctional planning tool. The analysis, designs, and actions must be based on the principles of sustainability and multifunctionality, and fulfil the requirements set out by the aforementioned Agency for the characterization of the green infrastructure elements.

The characterization of the periurban area and the subsequent assessment of its potential as a green infrastructure have generated three belts of connection around the principal landscape elements: periurban green areas, ecological corridors, agricultural areas, industrial and commercial areas, and linear infrastructures. The aims and actions to be developed in each case have produced five main guiding principles for action:

a.  Governance;
b.  Sustainable agriculture guidelines, for leisure and the local market;
c.  Landscape quality guidelines for industrial and commercial areas;
d.  Connectivity guidelines;
e.  Management guidelines for the urban and periurban green belt.

The characterization of the periurban space reveals the presence of elements of a high landscape value which, however, are unknown to the population and the local authority. Promoting them does not involve huge financial investment, but rather an emphasis on information and promotion of these sites by the authorities.

Large periurban parks play an essential role in the Open Space Strategy due to their social bonding and entrenchment, to their large area, to their location between large urban areas, and to their connection with ecological corridors. The weaknesses observed and the issues with their maintenance make them spaces with high potential as a laboratory for launching programs which are dynamic in design and can be ecologically maintained, thereby improving their integration in the surrounding landscape.

Where the urban–periurban border is concerned, it is concluded that, in the research area, industrial and commercial areas represent a physical, social, and ecological barrier making their landscape integration essential to achieving an environmental and functional connection. In relation to built-up areas and new developments, it is essential to design and maintain green areas as an alliance with the natural environment, instead of trying to fight against the incorporation of wildlife in our parks and gardens.

The design of this Open Space Strategy offers the authorities a cross-disciplinary planning tool which can be used at a landscape reading level. This comprehensive view enables the optimization of efforts and investments and involves the whole community at all levels of action through citizen participation.

## 5. Discussion

Throughout the research, the manner in which limits in the landscape are dealt with has been taken into account. Working on the urban–natural border presents the difficulty and the continuous challenge of devising strategies which penetrate the city and are determined by it. The limits of the field of research are focused on the periurban landscape, but references to the urban green fabric are inevitable for two reasons: the necessary connection with it in order to achieve sustainable strategies, and the lack of references on the subject.

This last aspect opens up multiple avenues of research and projects that have not been addressed because they are outside the scope of research, but are of great value, such as: the development of evaluation methodologies which allow the development of comparative studies of urban green spaces, the development of uniform characterization criteria that help to define green space management strategies, the implementation of pilot experiences relating to adaptive ecological design on different scales, or research into the application of the dynamics of the Mediterranean ecosystem in the design and maintenance of the green infrastructure.

**Author Contributions:** Conceptualization, E.F.-P.; methodology, E.F.-P., A.V.-V., Ó.L.-Z. and R.V.L.-D.; validation, E.F.-P. and A.V.-V.; formal analysis, Ó.L.-Z. and R.V.L.-D.; investigation, E.F.-P., A.V.-V., Ó.L.-Z., and R.V.L.-D.; writing—original draft preparation, E.F.-P., A.V.-V., Ó.L.-Z. and R.V.L.-D.; writing—review and editing, E.F.-P., A.V.-V., Ó.L.-Z. and R.V.L.-D.; visualization, E.F.-P. and A.V.-V.; supervision, Ó.L.-Z. and R.V.L.-D.; All authors have read and agreed to the published version of the manuscript.

**Funding:** This research received no external funding.

**Conflicts of Interest:** The authors declare no conflict of interest.

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
