# Peer review of "Periurban Areas in the Design of Supra-Municipal Strategies for Urban Green Infrastructures"

_forests, doi:10.3390/f12050626_

Round 1
Reviewer 1 Report
This article focuses on a very important topic: the planning of green infrastructures on a supramunicipal scale. The premise of exploring periurban green spaces as a green infrastructure promoting environmental, social and economic benefits is a very interesting approach. The results are particularly welcome, namely the detailed map of each strategy followed by the definition of each objective and the proposed actions.
Despite this very interesting approach, there are some important gaps in the presentation of the research design which ultimately affects the reading and the evaluation of the entire paper. For instance:
- the stated objective (l.37) doesn’t seem to fit neither the methods nor the achieved results.
- particularly, the organization of the “Methodology” section does not allow a comprehensive understanding of how the Supra-municipal Open Space Strategy was conceived and designed. For instance:
- Point 3.1. “Tools for supra-municipal assessment and planning” draws some concepts about the assessment of periurban green areas; but how was this assessment conducted?
- Point 3.2. “Research methodology” – the information provided about the eight categories which constituted the cartographic base for the design of the open space strategy is too vague.
- In the results section, it is presented a classification of the actions for each objective, carried out based on environmental, social and economic functions. How were these environmental, social and economic functions assessed?
Therefore, I would advise authors to reorganize the paper, particularly the Methodology section, in order to give a more comprehensive understanding of the research process.
Author Response
This article focuses on a very important topic: the planning of green infrastructures on a supramunicipal scale. The premise of exploring periurban green spaces as a green infrastructure promoting environmental, social and economic benefits is a very interesting approach. The results are particularly welcome, namely the detailed map of each strategy followed by the definition of each objective and the proposed actions.
Despite this very interesting approach, there are some important gaps in the presentation of the research design which ultimately affects the reading and the evaluation of the entire paper. For instance:
- the stated objective (l.37) doesn’t seem to fit neither the methods nor the achieved results.
Indeed, this expression doesn´t fit as the main objective of the research, so it has been eliminated from the text. The improvement of the quality of life of citizens is a consequence of the assessment of aspects related to accessibility, spatial design or sustainability criteria. It will be better explained in the methodology to highlight it. However, it´s removed from this paragraph of the summary so as not to imply that the article focuses on the valuation of social benefits.
- particularly, the organization of the “Methodology” section does not allow a comprehensive understanding of how the Supra-municipal Open Space Strategy was conceived and designed. For instance:
The methodology section has been reorganized and, for a better understanding, the methodological references used as a basis for the study are included.
- Point 3.1. “Tools for supra-municipal assessment and planning” draws some concepts about the assessment of periurban green areas; but how was this assessment conducted?
The evaluation of green areas has been carried out through a matrix of environmental, social and economic criteria. The process of defining these criteria is shown in Figure 1.
- Point 3.2. “Research methodology” – the information provided about the eight categories which constituted the cartographic base for the design of the open space strategy is too vague.
The definition of these categories has been carried out through the analysis of historical images, cartography, detailed field studies and bibliographic references such as the Municipal Management Plans. A text has been introduced to explain this process.
- In the results section, it is presented a classification of the actions for each objective, carried out based on environmental, social and economic functions. How were these environmental, social and economic functions assessed?
In response to this review, a paragraph and the figure 1 explaining this process have been included in the methodology section 3.2.
Therefore, I would advise authors to reorganize the paper, particularly the Methodology section, in order to give a more comprehensive understanding of the research process.

Reviewer 2 Report
General comments
This research deals with important aspects relative to establish creative processes which, through micro-scale activities (landscaping), generate the articulation of visible actions on a tecrritorial scale (landscape planning) in both the natural environment (environmental landscape planning) and the urban environment (town planning based on the landscape).
The findings show the below:
It is important to develop environmental quality standards to assess Green Infrastructures as a whole: the administrative processes, their design, construction, maintenance, and resilience. This research indicates how this change in the planning and management of green periurban areas improves citizens’ quality of life, boosts the multifunctionality of periurban spaces, improves the intrinsic quality of the landscape, promotes the city’s sustainability and resilience and improves governance. It should be noted that analysis, design, and action should be built on the premises of sustainability and multifunctionality and comply with the criteria for characterizing the elements as green infrastructure. In the field of study, the characterization of the periurban area and its subsequent assessment as a green infrastructure, provide the guidelines for action for devising an Open Space Strategy. This strategy constitutes a cross-disciplinary planning tool for local authorities when reading the landscape.
Specific Comments
- Please add recent references in all the MS. There are several paragraphs without references. Attention!
- Please moderate English changes required.
- Please read the journal instructions carefully and follow them strictly.
- Cross-reference all of the citations in the text with the references in the reference section.
- Make sure that all references have a corresponding citation within the text and vice versa.
- Double-check the spelling of the author names and dates and make sure they are correct and consistent with the citations.
- Spell out all journal titles in the reference section.
- Make sure that all figures and tables are cited within the text and they are cited in consecutive order.
- Make sure that all Figure captions are placed below the figures, while table captions must be placed above the Tables (Table 1, 2…).
- Please add all figures in better resolution.
- Please add section "Discussion"
- Please add section "Future Perspectives"
Author Response
Specific Comments
- Please add recent references in all the MS. There are several paragraphs without references. Attention!
Recent references have been added in several paragraphs of the MS.
- Please moderate English changes required.
The text has been proofread by a native speaker.
- Please read the journal instructions carefully and follow them strictly.
We have carefully read the instructions for authors and have incorporated a new "conflict of interest" section just before the bibliographic references.
- Cross-reference all of the citations in the text with the references in the reference section.
All references in the text have been crossed with the references in the reference section.
- Make sure that all references have a corresponding citation within the text and vice versa.
All references have been verified to have a corresponding citation within the text and vice versa.
- Double-check the spelling of the author names and dates and make sure they are correct and consistent with the citations.
Author names and dates have been checked and are consistent and correct with the citations.
- Spell out all journal titles in the reference section.
All the titles of the journals are written in full, without acronyms.
- Make sure that all figures and tables are cited within the text and they are cited in consecutive order.
We have checked that all tables and figures in the text have been cited in consecutive order. To facilitate the readability of the text, Figures 2 to 4 have been placed consecutively.
- Make sure that all Figure captions are placed below the figures, while table captions must be placed above the Tables (Table 1, 2…).
Since the text has been modified with the addition of three tables (tables 1, 2 and 3), the captions have been placed above as suggested by the reviewer.
- Please add all figures in better resolution.
Figures 2 to 6 have been modified in an attempt to improve their resolution.
- Please add section "Discussion"
A discussion section has been included with references to future perspectives, focused on the need to research work methodologies that help evaluate green infrastructure, not only in the design phase of new strategies, but also in the management and maintenance of the existing network.
- Please add section "Future Perspectives"
This revision has been included in the previous section.

Round 2
Reviewer 1 Report
The manuscript has been significantly improved and now warrants publication in Forests. Regards Helena MadureiraReviewer 2 Report
Dear Authors,
I consider the article is suitable for publication in its present form.